

# Calculating site-specific evolutionary rates at the amino-acid or codon level yields similar rate estimates

Dariya K. Sydykova and  Claus O. Wilke

Department of Integrative Biology, Center for Computational Biology and Bioinformatics, and Institute for Cellular and Molecular Biology, The University of Texas at Austin, Austin, TX, USA

## ABSTRACT

Site-specific evolutionary rates can be estimated from codon sequences or from amino-acid sequences. For codon sequences, the most popular methods use some variation of the $dN/dS$ ratio. For amino-acid sequences, one widely-used method is called Rate4Site, and it assigns a relative conservation score to each site in an alignment. How site-wise $dN/dS$ values relate to Rate4Site scores is not known. Here we elucidate the relationship between these two rate measurements. We simulate sequences with known $dN/dS$, using either $dN/dS$ models or mutation–selection models for simulation. We then infer Rate4Site scores on the simulated alignments, and we compare those scores to either true or inferred $dN/dS$ values on the same alignments. We find that Rate4Site scores generally correlate well with true $dN/dS$, and the correlation strengths increase in alignments with greater sequence divergence and more taxa. Moreover, Rate4Site scores correlate very well with inferred (as opposed to true) $dN/dS$ values, even for small alignments with little divergence. Finally, we verify this relationship between Rate4Site and $dN/dS$ in a variety of empirical datasets. We conclude that codon-level and amino-acid-level analysis frameworks are directly comparable and yield very similar inferences.

Corresponding author
Claus O. Wilke,
wilke@austin.utexas.edu

## INTRODUCTION

Different sites in a protein evolve at different rates (*Kimura & Ohta, 1974*; *Perutz, Kendrew & Watson, 1965*), and these rate differences are shaped by the interplay of functional and structural constraints each site experiences (*Echave, Spielman & Wilke, 2016*). For example, protein surface sites tend to evolve faster than interior sites of a protein (*Franzosa & Xia, 2009*; *Shahmoradi et al., 2014*; *Yeh et al., 2014b*; *Yeh et al., 2014a*; *Huang et al., 2014*; *Ramsey et al., 2011*; *Dean et al., 2002*; *Scherrer, Meyer & Wilke, 2012*; *Mirny & Shakhnovich, 1999*; *Zhou, Drummond & Wilke, 2008*). Active sites in enzymes tend to be highly conserved (*Jack et al., 2016*; *Dean et al., 2002*; *Kimura & Ohta, 1973*; *Huang et al., 2015*), and sites involved in protein–protein interactions are somewhat more conserved than other surface sites (*Mintseris & Weng, 2005*; *Kim et al., 2006*; *Franzosa & Xia, 2009*; *Jack et al., 2016*).

Analyses of sequence variation in a structural context frequently make use of site-specific evolutionary rate estimates, and a wide variety of different methods exist to infer such rates from either codon or amino-acid sequences (*Nielsen & Yang, 1998*; *Yang & Nielsen, 2002*; *Kosakovsky Pond, Frost & Muse, 2005*; *Kosakovsky Pond & Muse, 2005*; *Yang et al., 2000*; *Murrell et al., 2012*; *Lemey et al., 2012*; *Pupko et al., 2002*; *Fernandes & Atchley, 2008*; *Huang & Golding, 2014*; *Huang & Golding, 2015*; *Mayrose et al., 2004*). The most widely applied methods for codon sequences are based on $dN/dS$, the rate of non-synonymous substitutions per non-synonymous site $dN$ divided by the rate of synonymous substitutions per synonymous site $dS$. The $dN/dS$ ratio is commonly used to infer purifying ($dN/dS < 1$) or positive selection ($dN/dS > 1$) in protein-coding genes (*Nielsen & Yang, 1998*; *Goldman & Yang, 1994*). The most popular method for rate inference in amino-acid sequences is Rate4Site (*Pupko et al., 2002*; *Mayrose et al., 2004*). Rate4Site assigns a score to a site as a proxy for the rate of evolution at that site. Rate4Site is typically used to locate active sites on the protein structure, such as protein–protein interaction or protein–ligand interaction sites and catalytic sites (*Mousson et al., 2005*; *Fischer, Mayer & Söding, 2008*; *Tuncbag, Gursoy & Keskin, 2009*; *Bradford et al., 2006*; *Guney et al., 2008*). How $dN/dS$ inference methods relate to Rate4Site scores is not known.

The relationship between protein structure and evolutionary variation has been investigated in many different protein families and many different datasets of varying divergence levels and taxonomic origin, and some studies have used codon sequences and codon-related methods to infer the rate of evolution (*Franzosa & Xia, 2009*; *Shahmoradi et al., 2014*; *Scherrer, Meyer & Wilke, 2012*; *Zhou, Drummond & Wilke, 2008*; *Kim et al., 2006*) while others have used amino-acid sequences (*Ramsey et al., 2011*; *Yeh et al., 2014b*; *Yeh et al., 2014a*; *Huang et al., 2014*; *Huang et al., 2015*; *Jack et al., 2016*; *Mirny & Shakhnovich, 1999*). Because of these differences in datasets and analysis approaches, it is not obvious to what extent results from different studies can be compared. To the extent that different studies produce contradictory results, and they frequently do (*Jackson et al., 2016*; *Echave, Spielman & Wilke, 2016*), are these contradictions due to fundamental differences in the analyzed datasets (e.g., highly diverged sequences from many taxonomic groups vs. weakly diverged sequences from a single population) or in the employed methods to infer evolutionary rates (e.g., inference based on amino-acid sequences vs. on codon sequences)?

Here we address the second question, to what extent analyses at the codon level are comparable to analyses at the amino-acid level. Specifically, we use extensive simulations to ask how similar the site-specific Rate4Site scores are to site-specific $dN/dS$ values. We simulate sequence divergence both under $dN/dS$ models and under mutation–selection models, and we then ask how inferred Rate4Site scores for these simulated alignments compare to (i) the true simulated $dN/dS$ values at each site and (ii) the inferred $dN/dS$ values obtained from the simulated alignments. We find that Rate4Site scores generally correlate well with $dN/dS$, in particular if both quantities are inferred from sequence data. We verify this observation on rates inferred from empirical datasets, and we conclude that amino-acid level and codon-level analysis of rate variation will generally yield comparable results.

## METHODS

### Generation of simulated alignments

Our simulation approach was similar to the one employed by *Spielman, Wan & Wilke (2016)*. In brief, we first generated a set of balanced, binary trees with different branch lengths and numbers of taxa, using the R package ape (*Paradis, Claude & Strimmer, 2004*). We then simulated sequence evolution along these trees using the python library pyvolve (*Spielman & Wilke, 2015a*).

We generated a total of 40 trees, using all pairwise combinations of five different branch lengths and eight different numbers of taxa. The branch lengths we used were 0.0025, 0.01, 0.04, 0.16, and 0.64. These numbers indicate the divergence in mutations per site between two nodes in a tree. The numbers of taxa we used were 16, 32, 64, 128, 256, 512, 1,024, and 2,048.

To generate alignments with site-specific $dN/dS$ values, we simulated sequences with 100 codon sites using a site-specific Muse–Gaut model (*Muse & Gaut, 1994*). To simulate sequences with constant $dS$, we set $dS = 1$ at all sites and set $dN$ at each site to a different value randomly drawn from a uniform distribution between 0.1 and 1.6. To simulate sequences with variable $dS$, we assigned each site a distinct $dN$ and $dS$ value, by first choosing a randomly drawn $dN/dS$ value, then choosing a randomly drawn $dS$ value, and then setting $dN = dS \times (dN/dS)$. The $dN/dS$ values were drawn from a uniform distribution between 0.1 and 1.6, and the $dS$ values were drawn from a uniform distribution between 0.5 and 2. We generated 50 replicate sequence alignments for each combination of branch length, number of taxa (128–2,048), and choice of $dS$ (constant or variable), for a total of 2,500 sequence alignments.

We also generated alignments with gamma-distributed site-specific $dN/dS$ values. We simulated sequences with 100 codon sites using a site-specific Muse–Gaut model. We set $dN/dS$ to a randomly drawn value from a gamma distribution. We ran simulations for six distinct gamma distributions, using shape parameters $\alpha$ and rate parameters $\beta$ previously estimated for six HIV-1 proteins (Table 2 in *Meyer & Wilke, 2015b*). For each protein (i.e., distinct gamma distribution), we generated 50 replicate sequence alignments for each combination of branch length and number of taxa (16–256), for a total of 1,500 sequence alignments per protein.

For sequences simulated according to MutSel models, we used sequence alignments previously published in *Spielman, Wan & Wilke (2016)*, specifically the alignments simulated with unequal nucleotide frequencies. These sequences were simulated using the Halpern and Bruno model (HB98) (*Halpern & Bruno, 1998*), and we had alignments for the same tree parameters, $dS$ variation (constant/variable), and replicate numbers as our simulations of the $dN/dS$ model, for a total of 2,500 sequence alignments.

### Rate inference

To acquire the Rate4Site scores, the simulated sequences were translated into amino acids using biopython. The translated sequences were inputted into Rate4Site along with their

corresponding trees. We ran Rate4Site with the following options:

```
rate4site -s aln_file -t tree_file -o norm_rates_file \
        -y orig_rates_file
```

Here, `aln_file` is the input fasta file with aligned sequences. The file `tree_file` contains the phylogenetic tree. The file `norm_rates_file` is the output file into which Rate4Site writes $z$-normalized rate scores, and `orig_rates_file` is the output file into which Rate4Site writes original rate scores. The option `-y` causes Rate4Site to output original scores. (By default, Rate4Site only outputs $z$-transformed scores.) In our analysis we used only the original scores, renormalized such that they had a mean of 1.

We inferred site-specific $dN/dS$ using the one-parameter fixed-effects likelihood method (FEL1) implemented in HyPhy (*Kosakovsky Pond, Frost & Muse, 2005*). We ran HyPhy using the FEL1 script provided in *Spielman, Wan & Wilke (2016)*. After running the $dN/dS$ inference, we explicitly set $dN/dS = 0$ at all sites that did not experience any amino-acid changes. We did so because the FEL1 method assigns a site-wise $dN/dS$ of 1 to completely conserved sites that contain no synonymous and no non-synoymous mutations. However, $dN/dS$ should equal 0 at such sites in a one-parameter model, which implicitly assumes that $dS$ is the same at all sites and hence will be non-zero even at completely conserved sites.

## Analysis of empirical datasets

For analysis of empirical datasets, we used data from *Spielman & Wilke (2013)* and *Meyer & Wilke (2015b)*. From *Spielman & Wilke (2013)*, we acquired unaligned sequence data for six arbitrarily chosen membrane proteins: Mannose-6-phosphate receptor M6PR (Ensembl transcript ID: ENST00000000412), CD74 (Ensembl transcript ID: ENST00000009530), CD4 (Ensembl transcript ID: ENST00000011653), G protein-coupled receptor class C, GPRC5A (Ensembl transcript ID: ENST00000014914), Gamma-aminobutyric acid type A receptor, GABRA1 (Ensembl transcript ID: ENST00000023897), and TNF receptor superfamily member 17, TNFRSF17 (Ensembl transcript ID: ENST00000053243). We aligned the amino-acid sequences using MAFFT 7.305b (Multiple Alignment using Fast Fourier Transform) (*Katoh & Standley, 2013*). We ran MAFFT using default options with:

```
mafft input_fasta_file > output_fasta_file
```

Here, `input_fasta_file` is the fasta file containing sequences to be aligned and `output_fasta_file` is the output file into which the alignment is written. The aligned amino-acid sequences were subsequently back-translated to codon sequences using the unaligned nucleotide sequences.

For sequence data from *Meyer & Wilke (2015b)*, we used the aligned amino and nucleotide sequence files for all of the proteins used in the paper.

We inferred phylogenetic trees from amino-acid sequences using RAxML (*Stamatakis, 2014*). We ran RAxML with the following options:

```
raxmlHPC-PTHREADS-SSE3 -T 48 -s fasta_file -w output_directory \
        -n tree_name -m PROTCATLG -p 1
```

Here, `fasta_file` is the input file containing the aligned sequences. RAxML outputs all output files into the directory indicated by `output_directory`, and `tree_name` is the name for the output tree files. The option `-m PROTCATLG` picks a protein CAT model with LG matrix for the tree inference, and the option `-p 1` generates a random number seed for parsimony inference.

Finally, for all empirical datasets, we inferred Rate4Site scores and per-site $dN/dS$ values as described in the subsection "Rate inference."

## RESULTS

Both Rate4Site scores and per-site $dN/dS$ values are measures of the extent to which selection acts on individual protein sites. The Rate4Site model decomposes evolutionary distances in amino-acid alignments into a site-specific component $r_k$ and a branch-specific component $t_i$, such that the total divergence at site $k$ along branch $i$ can be written as $r_k t_i$. Here, $r_k$ is the Rate4Site score at site $k$ and $t_i$ is the branch length of branch $i$ in the phylogenetic tree. Importantly, $r_k$ is the same at all branches in the tree and $t_i$ is the same at all sites for each branch $i$. Because the rate decomposition is invariant under a rescaling of $r_k' = Cr_k$ and $t_i' = t_i/C$, Rate4Site scores are not unique unless an additional normalization condition is specified as well. The Rate4Site software solves this uniqueness problem by turning the $r_k$ into $z$-scores. However, the more natural normalization is to divide all $r_k$ by their mean, $r_k' = r_k/\sum_j r_j$, where the sum runs over all sites in the protein. These normalized $r_k'$ scores have the simple interpretation of providing the relative increase or decrease in substitution rate at site $k$ compared to the average rate of substitution in the rest of the protein.

In contrast to Rate4Site scores, which are calculated from amino-acid alignments, $dN/dS$ ratios are calculated on nucleotide alignments. They estimate the rate of non-synonymous divergence relative to the rate of synonymous divergence. However, just like in Rate4Site, in a site-specific $dN/dS$ model evolutionary divergence is decomposed into a site-specific $dN/dS$ value and a site-independent branch length. Thus, Rate4Site and site-specific $dN/dS$ measure fundamentally the same quantity. The main difference is the input data (amino-acid sequences vs. codon sequences) and the normalization (relative to mean across sites vs. relative to the synonymous divergence rate $dS$).

### Relationship between Rate4Site scores and true *dN/dS*

To determine the relationship between Rate4Site and $dN/dS$ models, we began by simulating sequence evolution with known, site-specific $dN/dS$ values and then comparing these true $dN/dS$ values to Rate4Site scores inferred from the simulated alignments (Fig. 1). We first considered the case of constant $dS$ among all sites. Thus, for each site in each alignment, we randomly drew a $dN$ from a uniform distribution ranging from 0.1 to 1.6. We set $dS = 1$ for all sites, such that the $dN/dS$ ratios similarly varied from 0.1 to 1.6. We ran simulations along a set of 25 balanced trees with different branch lengths and numbers of taxa, as used previously in a study of $dN/dS$ inference (*Spielman, Wan & Wilke, 2016*).

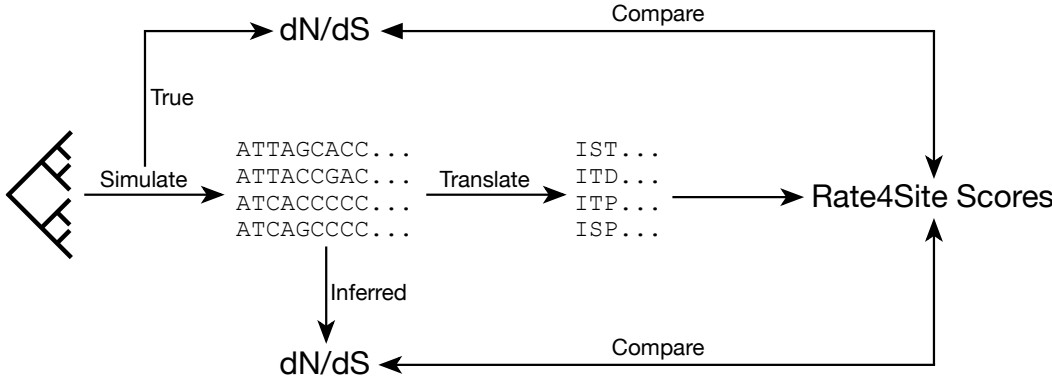

**Figure 1** **Analysis approach.** We assess the relationships between $dN/dS$ values and Rate4Site scores by simulating sequences with known $dN/dS$ values and then comparing either known or inferred $dN/dS$ values for these simulated alignments to the Rate4Site scores inferred on the same alignments.

Simulated sequences were 100 codon sites long, and we generated 50 replicate simulations for each simulation condition.

We calculated the correlation between each site's true $dN/dS$ and its inferred Rate4Site score to assess how well the Rate4Site scores agreed with the simulated rates. We then plotted the mean correlation strengths in replicate simulations against the simulations' branch lengths and number of taxa. We found that correlation strengths systematically increased with both increasing branch lengths and number of taxa (Fig. 2A). While correlations were low to moderate for the least-diverged and smallest alignments, for larger and/or more diverged alignments correlations approached values ranging from 0.8 to 1.0.

We also performed a comparison of the magnitude of Rate4Site scores and $dN/dS$ scores, by calculating root-mean-square deviations (RMSD) between these scores. Because these two types of scores are not measured in the same units, this comparison may not seem meaningful. However, we can convert both types of scores into normalized, relative scores by dividing them by their mean score. These normalized scores have comparable interpretations and RMSDs between them are meaningful quantities.

We found that RMSD values were generally moderate, between 0.1 and 0.6 (Fig. 2B). They declined with both increasing number of taxa and increasing sequence divergence. However, overall RMSD depended more strongly on branch length than on the number of taxa. Visual inspection of normalized Rate4Site scores plotted against normalized $dN/dS$ scores revealed no major systematic differences between these scores (Fig. 3). Differences seemed to be driven primarily by the sampling noise inherent in estimating site-specific evolutionary rates.

We repeated the same analysis but now using simulations in which $dS$ was allowed to vary among sites as well. The $dN/dS$ range was kept the same as before (0.1–1.6), but now each site had its own unique $dS$, randomly chosen from a uniform distribution ranging from 0.5 to 2. Overall, we found similar patterns in the variable $dS$ case as we had seen for constant $dS$ (compare Figs. 2C, 2D to Figs. 2A, 2B. However, correlations were generally somewhat weaker (Fig. 2C) and RMSD values somewhat higher (Fig. 2D) than what we had observed for constant $dS$. These results were to be expected, since Rate4Site as an

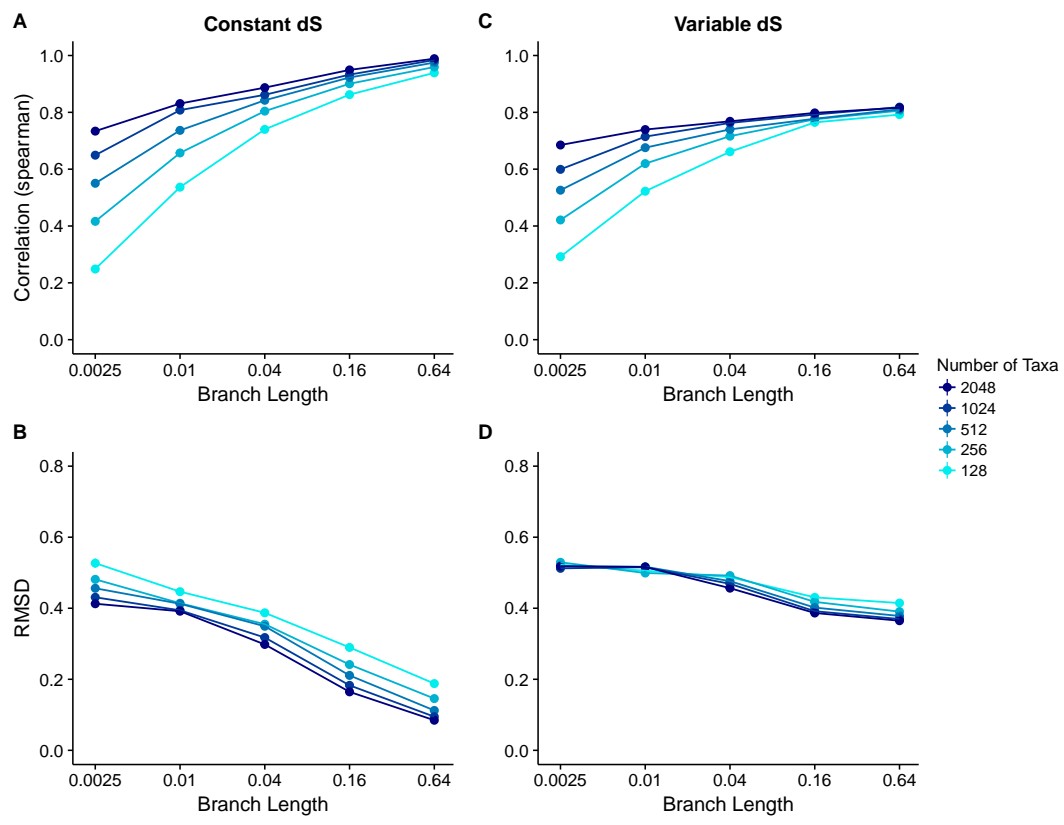

**Figure 2** **Relationship between Rate4Site scores and true site-specific $dN/dS$ for simulations performed with $dN/dS$ (Muse–Gaut) models.** Each point represents the mean over 50 replicate simulations. The error bars represent the standard error. In nearly all cases, error bars are smaller than the symbol size. (A) Correlations and (B) RMSD values between Rate4Site scores and true $dN/dS$, for the sequences simulated with constant $dS = 1$. (C) Correlations and (D) RMSD values between Rate4Site scores and true $dN/dS$, for the sequences simulated with variable $dS$.

amino-acid based metric does not take synonymous variation into account, and thus the $dS$ variation acts simply as added random noise on the $dN/dS$ scores compared to Rate4Site scores.

## Relationship between Rate4Site scores and scaled selection coefficients

The $dN/dS$ model is not a particularly realistic model of sequence evolution, because it does not have the notion of an underlying fitness landscape. A mutation increasing fitness should fix much more rapidly than the reverse mutation decreasing fitness. However, in a $dN/dS$ model, both mutations fix at the same rate. To increase realism in our analysis, we next investigated Rate4Site in the context of sequences simulated with mutation–selection (MutSel) models. MutSel models are specified by scaled selection coefficients, which describe the relative fitness of different amino acids (or codons) at each site in a sequence. We can derive expected $dN/dS$ values from these scaled selection coefficients (*Spielman & Wilke, 2015b*; *Dos Reis, 2015*) and hence we can ask how Rate4Site scores compare to the predicted $dN/dS$ values in MutSel models.

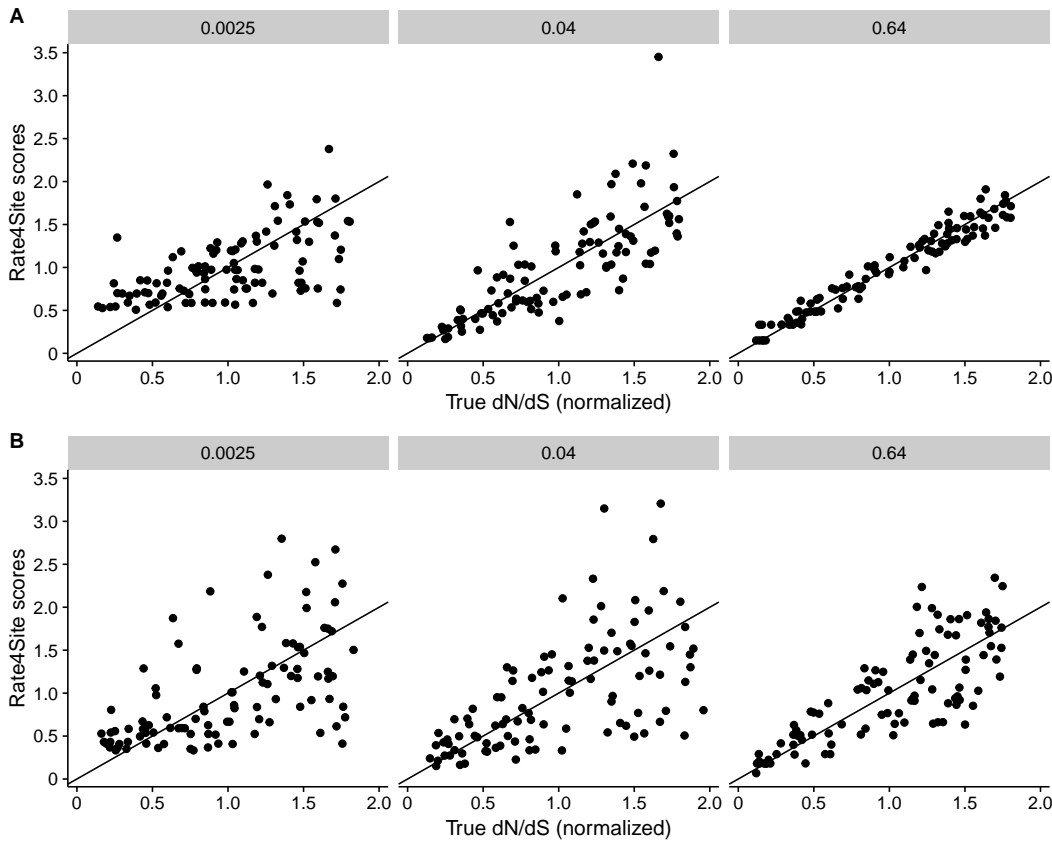

**Figure 3** **Rate4Site scores vs. true normalized** *dN/dS* **for a few example alignments simulated with a** *dN/dS* **model.** Each point represents one site in the simulated alignment, and the diagonal line represents the $x = y$ line. Numbers above each subplot indicate the branch length of the alignment, and the number of taxa was 512 in all cases. (A) Simulations with constant *dS* = 1. (B) Simulations with variable *dS*.

For this analysis, we employed previously published sequence alignments from *Spielman, Wan & Wilke (2016)*. These alignments had been simulated using the Halpern and Bruno model (HB98) (*Halpern & Bruno, 1998*) along the same 25 phylogenetic trees we employed in our previous analysis (five branch lengths in all pairwise combinations with five datasets with different numbers of taxa). As before, there were 50 replicates per simulation condition, and we again had one dataset with constant *dS* and one with variable *dS*. In the dataset with constant *dS*, all synonymous codons have the same fitness (neutral synonymous codons). In the dataset with variable *dS*, there are fitness differences among synonymous codons (non-neutral synonymous codons). See *Spielman, Wan & Wilke (2016)* for details of parameter choices.

Our results for sequences simulated with MutSel models were broadly similar to our results for sequences simulated with *dN/dS* models (Fig. 4). As before, correlations increased and approached 1 with increasing branch lengths and numbers of taxa, and RMSDs commensurately decreased. However, correlation strengths were consistently lower and RMSD values higher for the MutSel datasets than for the *dN/dS* datasets at the same sequence divergence and number of taxa (compare Figs. 4 to 2). As before,

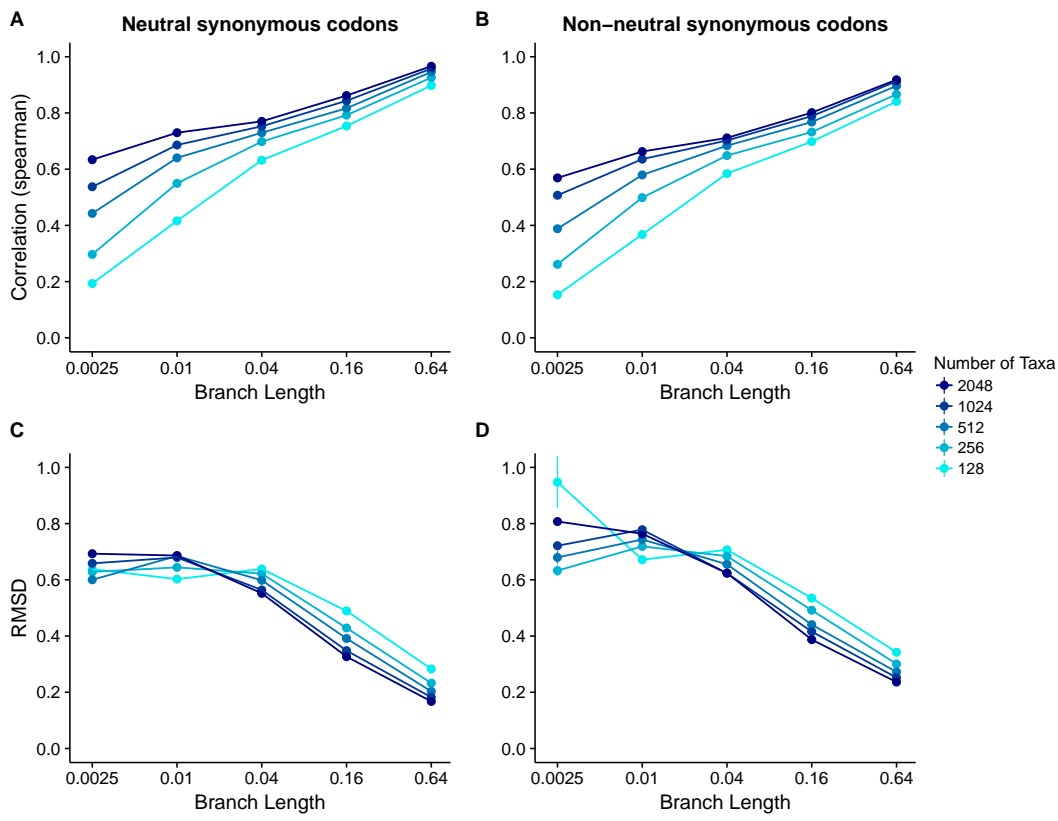

**Figure 4** **Relationship between Rate4Site scores and true site-specific *dN/dS* for simulations performed with MutSel (Halpern–Bruno) models.** Each point represents the mean over 50 replicate simulations. The error bars represent the standard error. In nearly all cases, error bars are smaller than the symbol size. (A) Correlations and (B) RMSD values between Rate4Site scores and true *dN/dS*, for the sequences simulated without codon bias (neutral synonymous codons). (C) Correlations and (D) RMSD values between Rate4Site scores and true *dN/dS*, for the sequences simulated with codon bias (non-neutral synonymous codons).

differences between normalized Rate4Site scores and normalized true *dN/dS* values seemed to be driven primarily by the sampling noise inherent in estimating site-specific evolutionary rates (Fig. 5). Finally, we found only minor differences between simulations with neutral synonymous codons and simulations with non-neutral synonymous codons. However, in the latter case, correlations were generally slightly lower and RMSD values somewhat higher (compare Figs. 4C and 4D to 4A and 4B).

## Relationship between Rate4Site scores and inferred *dN/dS*

The preceding analyses asked to what extent Rate4Site scores reflect the known underlying parameters used to generate the sequence alignments. An alternative question, possibly more applicable to practical sequence analysis, is to what extent Rate4Site scores mirror *dN/dS* values inferred on the same sequence data. To address this second question, we inferred site-wise *dN/dS* values for all sites in all alignments studied in the previous two subsections. The *dN/dS* values were inferred using the one-rate fixed-effects likelihood method (FEL1) implemented in HyPhy (*Kosakovsky Pond, Frost & Muse, 2005*). The FEL1

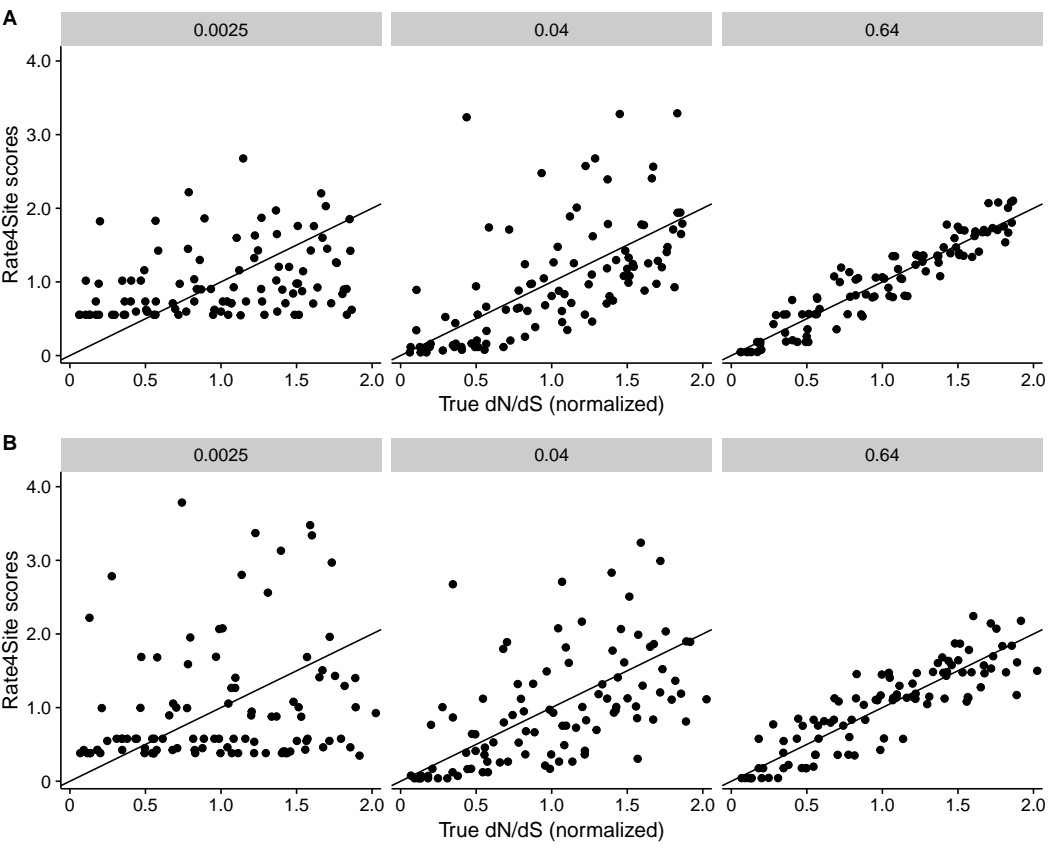

**Figure 5  Rate4Site scores vs. true normalized $dN/dS$ for a few example alignments simulated with a MutSel (Halpern–Bruno) models.** Each point represents one site in the simulated alignment, and the diagonal line represents the $x = y$ line. Numbers above each subplot indicate the branch length of the alignment, and the number of taxa was 512 in all cases. (A) Simulations without codon bias (neutral synonymous codons). (B) Simulations with with codon bias (non-neutral synonymous codons).

method assigns one $dN$ value per site and one $dS$ value across all sites in the sequence (*Spielman, Wan & Wilke, 2016*). Therefore, the variation in the inferred $dN/dS$ values is captured entirely by $dN$.

We found that Rate4Site scores were very highly correlated to inferred $dN/dS$ across all datasets and simulation conditions (Figs. 6 and 7). For sequences simulated with the $dN/dS$ model, correlations for all branch lengths exceeded 0.8 (Figs. 6A and 6C). Correlations generally increased and approached 1 for sequence alignments with more divergent sequences or more taxa. RMSD values were large for the smallest and least-diverged alignments but declined rapidly as either branch length or number of taxa increased (Figs. 6B and 6D). There was little difference between alignments simulated with constant $dS$ and with variable $dS$ (Figs. 6A vs. 6C and 6B vs. 6D). Results for sequences simulated with MutSel models were similar (Fig. 7). The main difference was that correlation coefficients were more sensitive to the number of taxa. For the lowest number of taxa (128) correlation coefficients were systematically lower when sequences were simulated with MutSel models rather than with $dN/dS$ models. The pattern reversed for the highest number of taxa

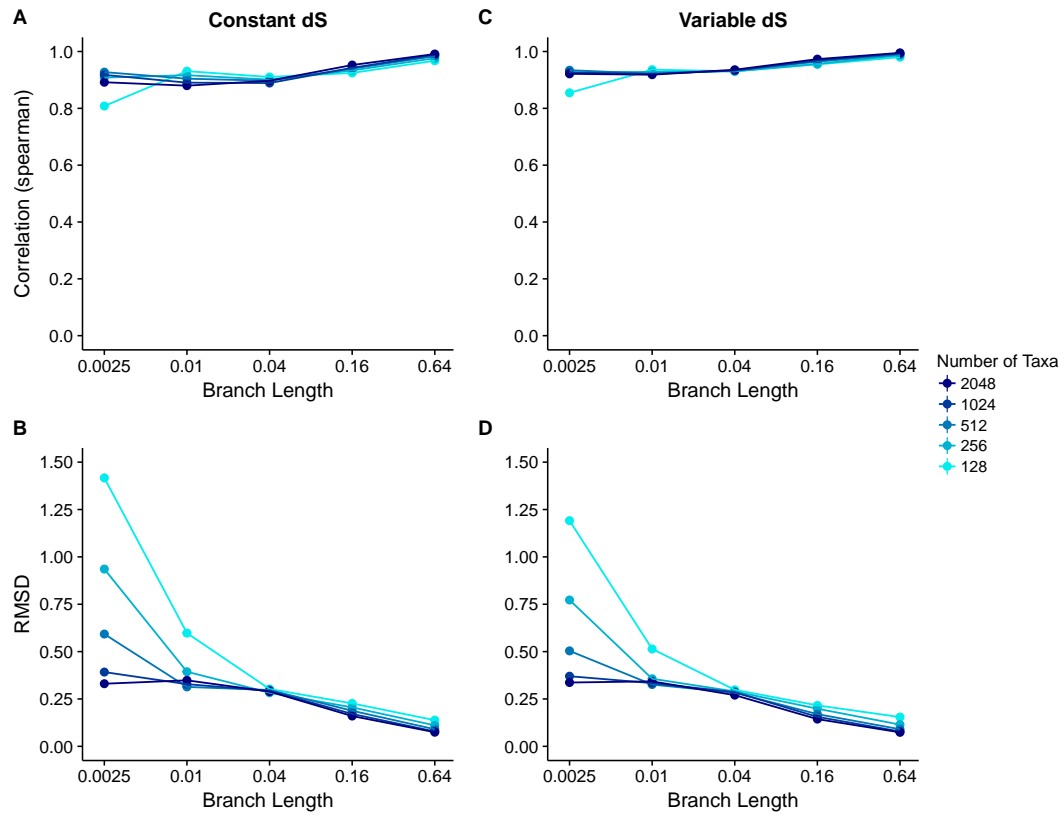

**Figure 6  Relationship between Rate4Site scores and inferred site-specific $dN/dS$ for simulations performed with $dN/dS$ (Muse–Gaut) models.** Each point represents the mean over 50 replicate simulations. The error bars represent the standard error. In nearly all cases, error bars are smaller than the symbol size. (A) Correlations and (B) RMSD values between Rate4Site scores and inferred $dN/dS$, for the sequences simulated with constant $dS = 1$. (C) Correlations and (D) RMSD values between Rate4Site scores and inferred $dN/dS$, for the sequences simulated with variable $dS$.

(2,048). RMSDs, however, were systematically higher for sequences simulated with MutSel models rather than with $dN/dS$ models. In all cases, there was little difference between sequences simulated with neutral synonymous codons and with non-neutral synonymous codons (Figs. 7A vs. 7D and 7B vs. 7D).

## Analysis of more realistic parameter settings and of empirical datasets

The above simulations utilized rates that were uniformly distributed across sites. In empirical datasets, however, the majority of sites evolves slowly and only very few sites evolve rapidly, with a resulting distribution that is approximately gamma (see e.g., *Meyer & Wilke, 2015b*). Therefore, we wanted to assess if the observed relationships between Rate4Site scores and $dN/dS$ hold for simulations with gamma-distributed true rates. We simulated sequences with gamma-distributed true $dN/dS$ values, using the shape and rate parameters obtained by *Meyer & Wilke (2015b)* for six different HIV-1 proteins. For these simulations, we also included a range of smaller numbers of taxa (16, 32, and 64) than before, to increase realism in the simulations.

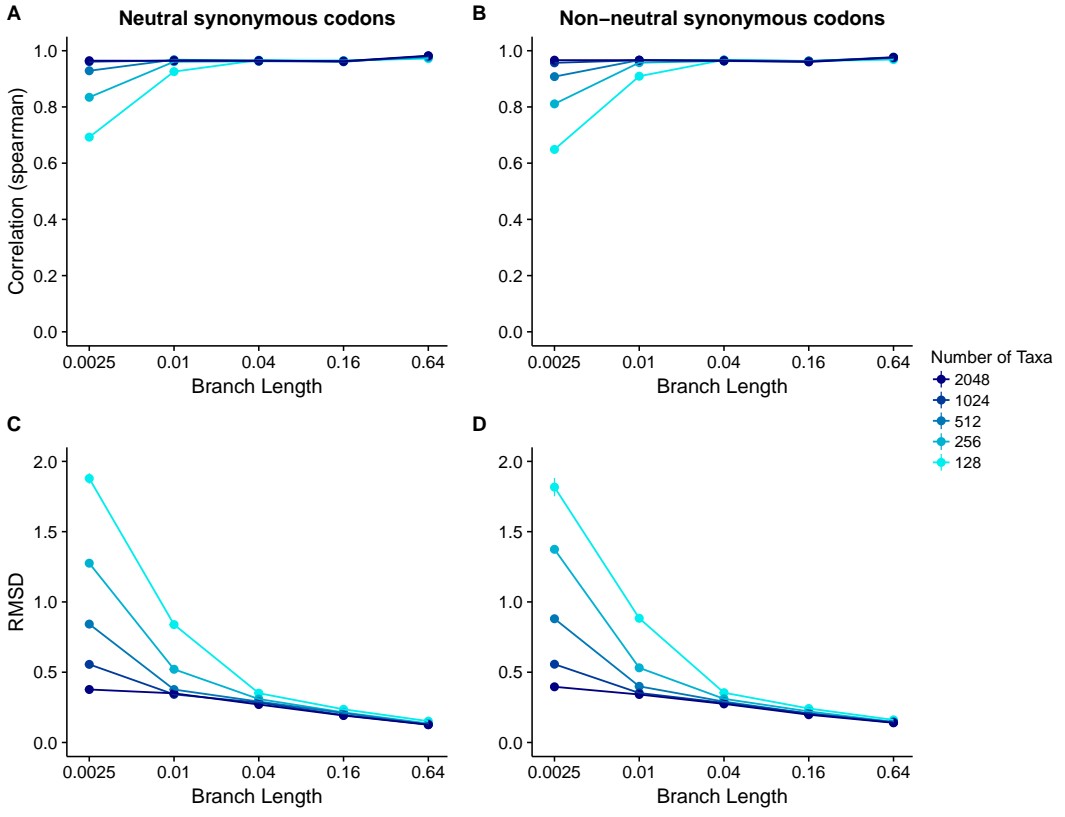

**Figure 7** **Relationship between Rate4Site scores and inferred site-specific $dN/dS$ for simulations performed with MutSel (Halpern–Bruno) models.** Each point represents the mean over 50 replicate simulations. The error bars represent the standard error. In nearly all cases, error bars are smaller than the symbol size. (A) Correlations and (B) RMSD values between Rate4Site scores and inferred $dN/dS$, for the sequences simulated without codon bias (neutral synonymous codons). (C) Correlations and (D) RMSD values between Rate4Site scores and inferred $dN/dS$, for the sequences simulated with codon bias (non-neutral synonymous codons).

We found a wide range of correlations between Rate4Site and true $dN/dS$ (Fig. 8A and Parts A of Figs. S1–S5), but the overall pattern was similar to what we had previously observed for uniformly distributed rates: Correlations increased and approached 1 as either sequence divergence or the number of taxa increased. Also note that the lowest number of taxa now was 16 vs. 128 in Fig. 2, and thus at identical numbers of taxa and branch lengths, the correlations for gamma-distributed rates were generally higher than the correlations for uniformly distributed rates. On the other hand, the RMSD values between Rate4Site scores and true $dN/dS$ were higher than in our previous analysis (compare Figs. 8B to 2B).

We also inferred $dN/dS$ from the simulated sequences and compared them to the Rate4Site scores. Unlike in our previous analysis on inferred $dN/dS$ (Fig. 6), the correlations between Rate4Site and inferred $dN/dS$ now spanned a wide range of values (Figs. 8C and Parts C of Figs. S1–S5) but were generally higher than the corresponding correlations between Rate4Site and true $dN/dS$. On the flip side, RMSD values were larger when calculated for inferred $dN/dS$ than when calculated for true $dN/dS$, and there was a notable uptick in RMSD at the highest branch lengths (Figs. 8D and Parts D of

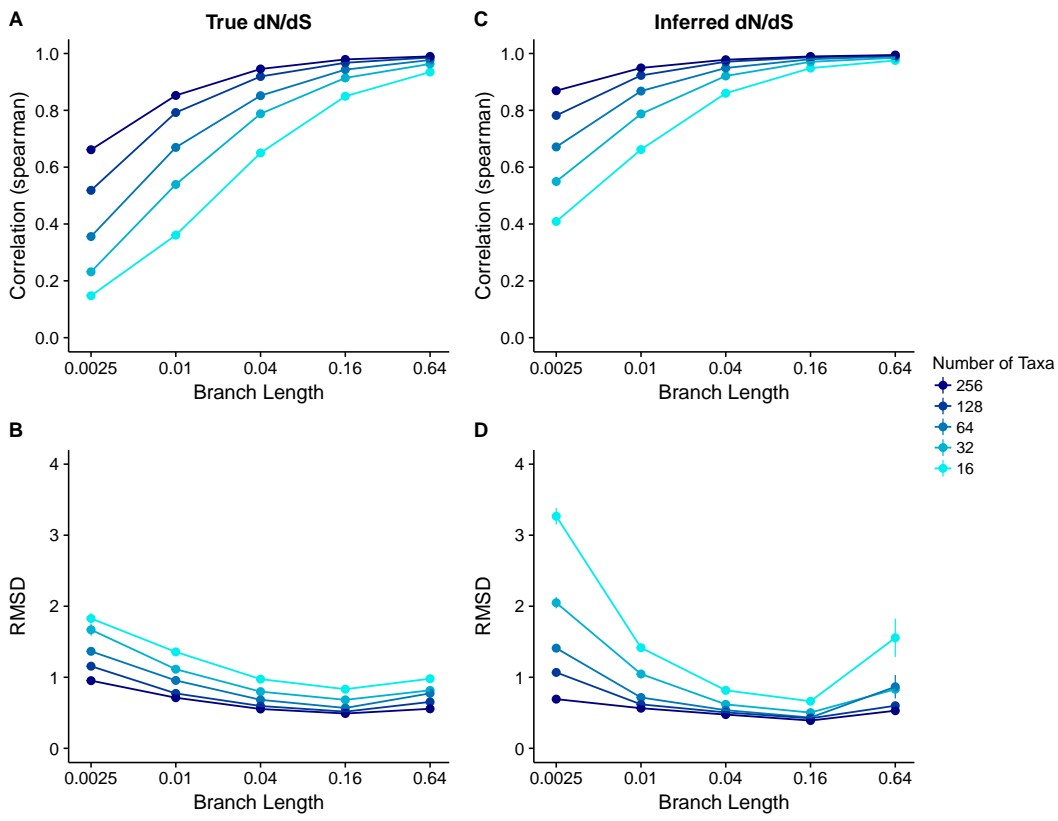

**Figure 8** **Relationship between Rate4Site scores and site-specific *dN/dS* for simulations performed with a *dN/dS* model with gamma-distributed true rates.** We used the gamma distribution observed for the HIV-1 integrase protein, with shape parameter $\alpha = 0.312$ and rate parameter $\beta = 1.027$ (*Meyer & Wilke, 2015b*). Each point represents the mean over 50 replicate simulations. The error bars represent the standard error. In nearly all cases, error bars are smaller than the symbol size. (A) Correlations and (B) RMSD values between Rate4Site scores and true *dN/dS*. (C) Correlations and (D) RMSD values between Rate4Site scores and inferred *dN/dS*. All simulations were performed without codon bias (neutral synonymous codons).

Figs. S1–S5). These observations can be explained with imprecise point estimates produced by both Rate4Site and *dN/dS* inference at both low and high sequence divergence: In general, both methods correctly assess which sites in an alignment evolve more rapidly than which other sites, and hence correlations are high, but the specific rate estimates assigned to individual sites are poor at both low and high sequence divergence, and hence RMSDs are high as well.

Finally, we asked to what extent the results found for simulated sequences carry over to empirical datasets. We inferred both Rate4Site scores and site-wise *dN/dS* in two distinct datasets, one consisting of six membrane proteins in mammals (taken from *Spielman & Wilke, 2013*) and one consisting of six alignments of HIV-1 genes (taken from *Meyer & Wilke, 2015b*). For both datasets, we measured the correlation between the original (non-normalized) *dN/dS* and normalized Rate4Sites scores. We found that Rate4Site scores and *dN/dS* were highly correlated, with correlation coefficients exceeding 0.8 in all

cases (Fig. 9). Thus, we can conclude that our simulation results carry over to empirical datasets, and that Rate4Site scores are generally comparable to inferred $dN/dS$ values.

Note that we plotted raw (non-normalized) $dN/dS$ in Fig. 9, and as a result points are not expected to fall onto the $x = y$ line. We plotted the data in this way because $dN/dS$ values are not typically normalized to their mean, and we wanted to assess how similar in magnitude raw $dN/dS$ scores are to Rate4Site scores. We found that for proteins under strong positive selection (such as HIV-1 gp120), raw $dN/dS$ is virtually identical to Rate4Site. However, for more typical proteins that experience little positive selection, raw $dN/dS$ values tend to be up to a factor 10 smaller than the corresponding Rate4Site scores.

## DISCUSSION

We have compared codon-level site-specific evolutionary rates estimated via $dN/dS$ to amino-acid level site-specific rates estimated via Rate4Site. We have found that Rate4Site scores correlate well with the known true $dN/dS$ values both in sequences simulated with a $dN/dS$ model and in sequences simulated with a MutSel model. Correlations generally increase and approach 1 for more diverged sequence alignments and for alignments with more taxa. Correlations are generally somewhat stronger when there is no variation in $dS$ among sites, though this effect is minor. We have also compared Rate4Site scores to inferred $dN/dS$ values and have found high correlations between the two measures, even for less diverged and smaller alignments. Finally, we have verified the relationship between Rate4Site scores and $dN/dS$ in a set of empirical datasets, and have found correlations close to 1 between the two rate estimates.

Surprisingly, even in scenarios of low sequence divergence or few taxa, when Rate4Site scores are only weakly correlated with the true $dN/dS$, we have found that they nevertheless correlate highly with inferred $dN/dS$. For all levels of sequence divergence and numbers of taxa, Rate4Site scores always correlate more strongly with inferred $dN/dS$ than with true $dN/dS$. While this observation may seem counterintuitive, there is a simple explanation. The evolutionary process is stochastic, and by random chance some sites will experience substantially more or fewer mutations than expected given their true rate. At those sites, all rate inference methods are expected to over/under-estimate the true rate, since their only input are the mutations that actually happened. As a result, we would generally expect different inference methods to produce results that are more similar to each other than they are to the true, underlying parameters. In agreement with this reasoning, a strong relationship between Rate4Site and inferred $dN/dS$ was also evident in empirical datasets. For the majority of alignments that we considered, correlation coefficients were in excess of 0.9. For the two alignments with the lowest correlation coefficients, of 0.83 for both alignments, the number of sequences were 19 and 22 and sequence divergence was low. Thus, unless alignments are very small and/or have very little divergence, Rate4Site scores and site-specific $dN/dS$ can be expected to correlate strongly in all cases. These findings demonstrate that Rate4Site and $dN/dS$ approaches have comparable ability to infer rates from sequence alignments. For sufficiently diverged alignments, both methods

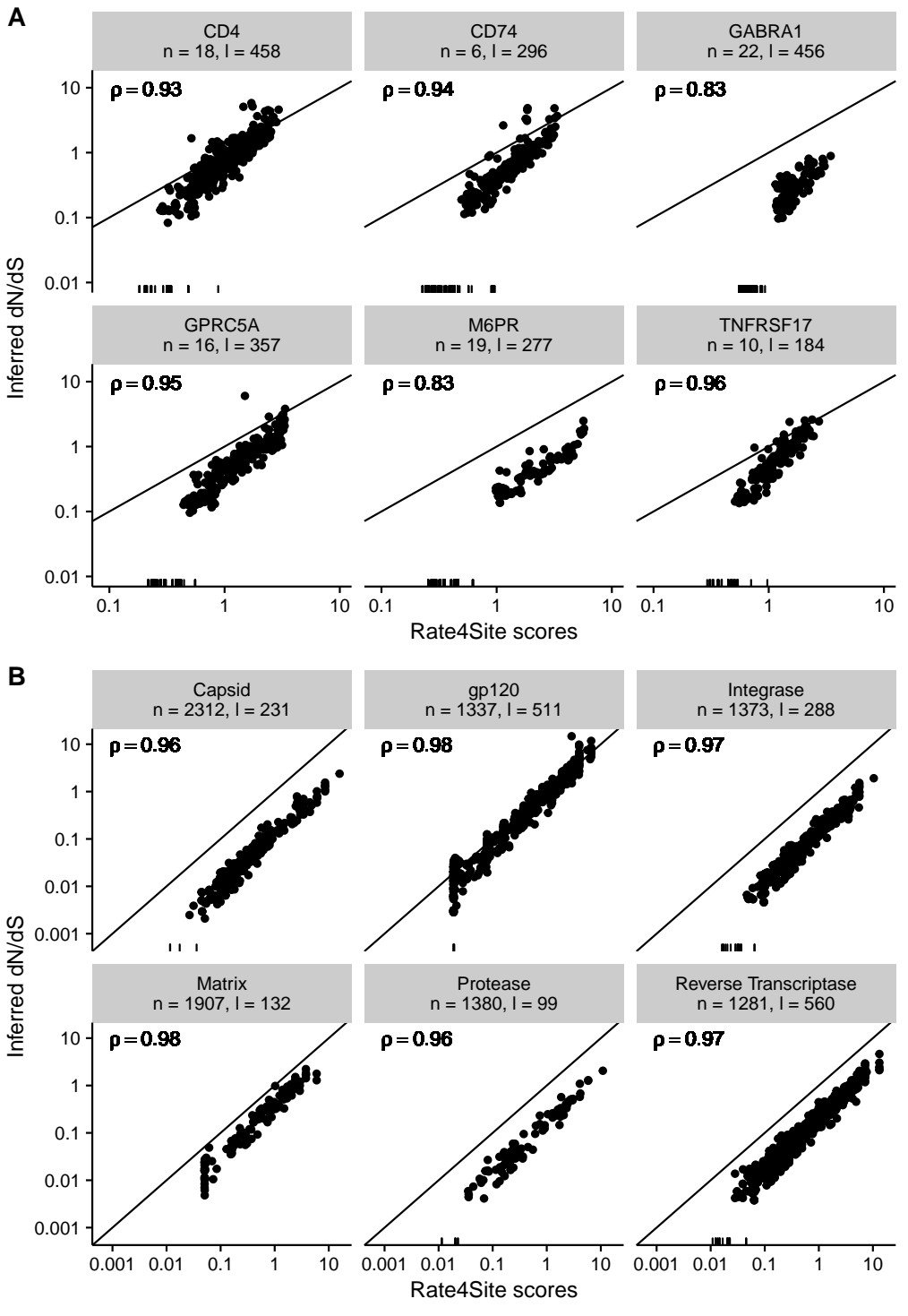

**Figure 9** **Inferred _dN/dS_ vs. Rate4Site scores for empirical datasets.** Each dot represents one site in the respective alignment, and the diagonal line represents the $x = y$ line. Rugs along the $x$-axis show sites with $dN/dS < 0.001$. Correlation coefficients are Spearman $\rho$, and all correlations are significant ($p < 10^{-15}$ throughout). Note that $dN/dS$ values were not normalized to a mean of one here, unlike Figs. 3 and 5. (A) Inferred $dN/dS$ vs. Rate4Site scores for six membrane proteins taken from _Spielman & Wilke (2013)_. (B) Inferred $dN/dS$ vs. Rate4Site scores for six HIV-1 proteins taken from _Meyer & Wilke (2015b)_.

accurately recover the true underlying rates. And for alignments with less divergence, they mis-estimate the underlying rates in a similar fashion.

In several cases, we saw high RMSDs despite high correlations between Rate4Site and inferred $dN/dS$. This observation suggests that both approaches rank sites similarly in terms of whether they are more rapidly or more slowly evolving, even when the point estimates for rates differ. One major factor causing differences in point estimates is the specific statistical approach used for rate inference in Rate4Site and in our $dN/dS$ estimation. In particular, Rate4Site uses a random-effects likelihood (REL) approach in a Bayesian framework, while for $dN/dS$ we employed a fixed-effects likelihood (FEL) approach in a maximum-likelihood framework. Prior work has shown that REL and FEL inference for the same model (e.g., $dN/dS$) tends to produce comparable estimates (*Meyer, Dawson & Wilke, 2013*; *Kosakovsky Pond & Frost, 2005*). However, the dynamic range for REL models tends to be reduced relative to that of FEL models, and in particular, REL models tend to overestimate the rates of sites with no or very few mutations. This pattern is evident for example in the left-most panel of Fig. 3A, where the Rate4Site scores do not fall below 0.5. Because of the tendency of REL models to overestimate rates of completely conserved sites, we generally prefer FEL approaches for site-specific rate estimation, and we believe that an FEL version of the Rate4Site model would be a useful addition to the toolbox of rate-inference approaches.

The $dN/dS$ metric is frequently used to identify sites under positive selection in viruses (*Vijaykrishna et al., 2008*; *Wood et al., 2009*; *Demogines et al., 2013*; *Meyer & Wilke, 2015a*). By contrast, Rate4Site has been mostly applied to identify conserved sites that correspond to protein–protein interaction sites or active sites in enzyme (*Mousson et al., 2005*; *Fischer, Mayer & Söding, 2008*; *Tuncbag, Gursoy & Keskin, 2009*; *Bradford et al., 2006*; *Guney et al., 2008*). Our results here show that for purposes of finding the most conserved or most rapidly varying sites in a sequence alignments, both methods would likely identify similar sites. One advantage of the $dN/dS$ approach, of course, is the ability to test whether $dN/dS$ is significantly above 1. When using Rate4Site scores, one can identify the most rapidly varying sites but one cannot run a statistical test that would determine whether the site is positively selected or not.

Recently, there has been considerable interest in linking site-specific rate variation to structural features of proteins (*Echave, Spielman & Wilke, 2016*). Studies addressing this topic have considered both $dN/dS$-based methods (*Scherrer, Meyer & Wilke, 2012*; *Franzosa & Xia, 2009*; *Shahmoradi et al., 2014*; *Kim et al., 2006*; *Meyer & Wilke, 2015b*; *Meyer & Wilke, 2015a*) and Rate4Site scores (*Huang et al., 2014*; *Yeh et al., 2014b*; *Yeh et al., 2014a*; *Jack et al., 2016*; *Huang et al., 2015*), though these studies have generally been done on disparate datasets. The extent to which results found with $dN/dS$ carry over to Rate4Site and vice versa has not been clear. Our findings here show that since the two methods infer rates that correlate strongly with each other, either type of inferred rate should produce comparable correlation strengths with structural features such as solvent accessibility.

We note several caveats to our conclusions. First, our simulated alignments were generally large and diverged, even for the smallest number of taxa and lowest branch

lengths. Even smaller and/or less diverged alignments will yield more noisy, less reliable Rate4Site inferences. Second, all our simulated alignments were obtained under the assumption that sites evolve independently from each other and that the rate of evolution does not change over time. These assumptions will generally increase the congruence between the true, simulated $dN/dS$ and the inferred Rate4Site score. However, the strong correlations we observed between Rate4Site scores and $dN/dS$ in several empirical datasets demonstrate that neither of these assumptions and limitations fundamentally invalidate our main findings. Amino-acid level and codon-level analyses of sequence data will generally yield comparable estimates of site-specific rates of evolution.

### Funding

This work was supported by National Science Foundation Cooperative agreement no. DBI-0939454 (BEACON Center), National Institutes of Health Grants R01 GM088344, and Army Research Office Grant W911NF-12-1-0390. The funders had no role in study design, data collection and analysis, decision to publish, or preparation of the manuscript.

### Grant Disclosures

The following grant information was disclosed by the authors:
National Science Foundation Cooperative agreement: DBI-0939454.
National Institutes of Health Grants: R01 GM088344.
Army Research Office Grant: W911NF-12-1-0390.

### Competing Interests

Claus O. Wilke is an Academic Editor for PeerJ.

### Author Contributions

- Dariya K. Sydykova conceived and designed the experiments, performed the experiments, analyzed the data, contributed reagents/materials/analysis tools, wrote the paper, prepared figures and/or tables, reviewed drafts of the paper.
- Claus O. Wilke conceived and designed the experiments, wrote the paper, reviewed drafts of the paper.

### Data Availability

All data and code are available on Github:
https://github.com/wilkelab/r4s_benchmark.

### Supplemental Information

Supplemental information for this article can be found online at http://dx.doi.org/10.7717/peerj.3391#supplemental-information.

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
