# Peer review of "Calculating site-specific evolutionary rates at the amino-acid or codon level yields similar rate estimates"

_PeerJ, doi:10.7717/peerj.3391_

## Round 0.1 · original submission · Minor Revisions

In this manuscript the authors use simulation studies to compare site-specific evolutionary rates calculated at the amino acid (Rate4Site) or nucleotide levels (dN/dS or mutation-selection models). The conclusion is that rates inferred at these two levels are strongly correlated. They then make a comparison with actual sequences from two empirical data sets, again concluding that the two measures are strongly correlated. The two external reviewers found the manuscript to be well-written, that the conclusions are supported by analyses, and that it represents a useful contribution. I agree and a suggest that the manuscript be accepted pending the minor revisions suggested in the constructive external reviews. Please respond to each of the reviewer's comments point-by-point in the cover letter of the revised manuscript.

A couple minor editorial suggestions below:

line 20: the correlation strengths increase in alignments with greater sequence divergence and more taxa
line 69: Here and elsewhere the phrase "natural sequence alignments" seems awkward to me. I know what you mean of course, but it conjures up this image in my mind of sequence alignments roaming the plains. How about actual sequences, real sequences, or empirical data sets?
line 167: Correlations generally increase and approach 1 for sequence alignments with more divergent sequences or more taxa.

Reviewer 1 ·

Basic reporting

Statistical phylogenetic models have been widely used to quantify natural selection acting on protein sequences. In this work, the authors studied the relationship between two types of substitution models, i.e., codon and amino acid substitution models, in the context of inferring purifying selection acting on proteins. It is well known that many amino acid substitution models, e.g., the PAM1 model, largely reflect codon structures. Therefore, it is not surprising that the codon and amino acid level models often generate similar results. Overall, this paper is well written and provides insights on the relationship between different substitution models. Background information is well described in this paper and the structure of the paper conforms to PeerJ standards.

Experimental design

This paper mainly used simulations to study the relationship between codon and amino acid substitution models. The authors investigated how branch lengths, numbers of taxa, and presence/absence of dS variability influence the estimation of substitution rates. Overall, the analyses are well designed and the results are solid.

A limitation of the current manuscript is that the authors have not extensively explored how the distribution of dN/dS ratios influence the results. For example, the authors assumed that dN/dS ratios follow a uniform distribution ranging from 0.1 to 1.6. However, in real data, substitution rates across sites are better described by the gamma distribution and the shape parameter in the gamma distribution is typically smaller than 1, suggesting that a large proportion of sites evolve at low rates and only a small fraction of sites evolve at high rates. The skewness of substitution rates towards lower values is not captured by the simulations in this paper. Therefore, I suggest the authors to perform additional simulations under the assumption that rate variability follows the gamma distribution and investigate how the shape parameter in the gamma distribution influences the performance of different models.

Validity of the findings

The main conclusion of this paper is that phylogenetic inferences of substitution rates based on codon and amino acid level models often yield similar results. This conclusion is intuitive and well supported by the analyses in this paper.

·

Basic reporting

The writing is very good.

Experimental design

The design is adequate to the job.

Validity of the findings

The conclusions are supported by the results.

Additional comments

Comments on "Calculating site-specific evolutionary rates at the amino acid
or codon level yields similar rate estimates"
(PeerJ) by Sydykova and Wilke.

This paper compares the estimates of evolutionary rates obtained depending on
whether on looks at the codon level or at the amino acid level.

I feel that the paper is clear and unambiguous. The paper is well written
and, in my opinion the conclusions are supported by the data presented. The
figures and tables are clear and well presented. The methodology is well
described and the background material is suitable.

So on the whole I think that the manuscript is well done and should be
accepted for publication. I have only minor comments about the manuscript and
think that they should be optional for the authors -- for their consideration
only as things to improve.

First, I find that the statements in the abstract are somewhat, at
first glance contradictory. "We find that Rate4Site scores generally
correlate well with true dN/dS and the correlation strengths increase in
alignments with higher sequence divergence ... Moreover ... [they are
nearly perfect] ... with little divergence." The correlation cannot
start off as increasing to a reasonable level with high divergence and
then decrease to nearly perfect.

The statement on line 57 that different studies frequently produce
contradictory results requires a reference.

I thought that the explanation on lines 84 to 90 was well done.

The statement on line 139 was first read by me as meaning that you
switched to data with "five ... taxa". The authors might wish to convert
this to "five sets of taxa numbers" or better "five data sets with different
numbers of taxa".

On lines 178-179 change "several" to "six". You can be precise.

On lines 190, 192 the use of "nearly perfect correlations" is a poor choice of
words. What is nearly perfect.

On line 201, you evaluate data sets with 19 and 22 taxa. Such small numbers
were never examined by the simulations. The simulations do show however, that
the results get worse with smaller data sets (not surprisingly). So why not
do the simulations in the range of the size of data sets that people (and
yourselves) actually use. Even the smallest simulation data set had 6 times
more taxa.

I am curious, from figure 2, why there is a higher correlation with a small
number of taxa with variable dS as opposed to constant dS at low branch length.

Again, for figure 6, inferring two variables gives a better correlation than
inferring one and comparing to the known value?

For figure 8, the number of taxa being considered and the length of the
alignment being used can usefully be annotated in the titles of each panel
(n=?, l=?).

For figure 8, the results seem to be biased. This is an obvious feature of
the figure but is not discussed.

---

## Round 0.2 · accepted · Accept

You have done a very nice job of responding to the reviewers comments and suggestions. I appreciate the organized response and think the additional analyses have extended and strengthened the conclusions of the paper.